# Exploring Rural Adolescents’ Dietary Diversity and Its Socioeconomic Correlates: A Cross-Sectional Study from Matlab, Bangladesh

**DOI:** 10.3390/nu12082230

**Published:** 2020-07-26

**Authors:** Mohammad Redwanul Islam, Syed Moshfiqur Rahman, Chandan Tarafder, Md. Monjur Rahman, Anisur Rahman, Eva-Charlotte Ekström

**Affiliations:** 1Department of Women’s and Children’s Health, Uppsala University, SE-751 85 Uppsala, Sweden; syed.moshfiqur@kbh.uu.se (S.M.R.); lotta.ekstrom@kbh.uu.se (E.-C.E.); 2International Center for Diarrheal Disease Research, Bangladesh (ICDDR,B), Dhaka 1212, Bangladesh; cktarafder@gmail.com (C.T.); monjur@icddrb.org (M.M.R.); arahman@icddrb.org (A.R.)

**Keywords:** dietary diversity, dietary pattern, rural adolescents, adolescent nutrition, household food security, Bangladesh

## Abstract

The majority of 36 million Bangladeshi adolescents live in rural areas. Improved understanding of their dietary patterns is of great public health importance. This study aimed to explore dietary diversity (DD) with its socioeconomic and gender stratification in a rural adolescent cohort and to isolate factors associated with inadequate DD. Household survey provided data for constructing dietary diversity scores (DDS) and assessing relevant socio-demographic variables. Final analysis included 2463 adolescents. Means and proportions were compared, and a binary logistic regression model was fitted. Inadequate DD was observed among 42.3% (40.3–44.2). Consumption of nutrient-rich foods varied significantly across gender and SES categories. Belonging to the poorest households (adjusted odds ratio (aOR) 1.59; 95% CI: 1.27, 2.00) and food insecure households (aOR 1.34; 95% CI: 1.13, 1.59), adolescents’ attainment of secondary education (aOR 1.38; 95% CI: 1.11, 1.71), and having mothers with secondary education or above (aOR 0.76; 95% CI: 0.60, 0.96) were associated with inadequate DD. Compared with girls from food secure households, girls from food insecure ones had higher odds of inadequate DD (aOR**_girl_** 1.42; 95% CI: 1.12, 1.81). Improving rural adolescents’ DD would require targeted interventions as well as broader poverty alleviation.

## 1. Introduction

Adolescence is a critical phase in the life course, characterized by remarkable physical growth, physiological, and cognitive development, and biological maturation [1,2]. It is during this phase that there occurs attainment of 40–60% of the peak bone mass and up to 50% of adult body weight along with a 15–20% increase in height [3]. Higher physiological demand for macro- and micro-nutrients [4] underlies the nutritional vulnerability of adolescents, which is profound in low- and middle-income countries (LMICs) [5]. Moreover, adolescence presents an additional “window of opportunity” to correct the nutritional deficits and growth faltering initiated during the first decade of life [6,7]. This is of great importance for LMICs, like Bangladesh, where still a significant proportion of children enter adolescence being stunted and/or underweight [5,8]. Therefore, intake of a diversified diet rich in essential nutrients is pivotal in resolving these adolescents’ nutritional vulnerability and in tackling different forms of malnutrition among them [9].

Bangladesh is home to around 36 million adolescents, forming 21% of its population, the majority of which live in rural areas [10]. Optimum adolescent nutrition is critical for Bangladesh to benefit from this huge potential workforce. Although the country recorded impressive reductions in childhood undernutrition [11], limited published estimates reveal high prevalence of stunting and underweight among Bangladeshi adolescents with girls suffering more [5,12,13]. Widespread micronutrient deficiencies have also been captured among adolescents [14]. On the other hand, overweight and obesity among adolescents have started to increase in Bangladesh [15]. Hence, adolescents’ consumption of a diversified diet that promotes a balanced delivery and adequacy of nutrients merits priority if Bangladesh is to accelerate progress toward the Global Goals [16].

Dietary diversity (DD) serves as a proxy for nutrient adequacy by indicating the extent of consumption of diverse foods across and within a number of food groups over a given reference period [17]. A diversified diet is assumed to reduce the likelihood of developing deficiency or excess of any particular nutrient [18]. Validation studies from LMICs document a persistent, positive association between higher DD and nutrient adequacy [19,20,21,22]. Nevertheless, little is known about rural adolescents’ DD in Bangladesh and their consumption pattern of foods from different food groups. Previous studies largely focused on the DD of under-5 children, pregnant adolescents, and women of reproductive age, with the latter two categories including a proportion of adolescent girls [13,23,24,25,26,27,28]. This study aimed to explore DD in a cohort of rural adolescents from Matlab, Bangladesh, with a systematic focus on the underlying socioeconomic correlates. Identification of these correlates would better inform policies on adolescent health and contribute to the development of targeted interventions. Our objectives were to: (i) describe and analyze DD and consumption pattern of foods from different groups along with their social and economic stratification, and (ii) identify socioeconomic and demographic predictors of inadequate DD among these adolescents.

## 2. Materials and Methods

### 2.1. Study Design, Participants, and Setting

This cross-sectional study was nested in the larger project of 15-year follow-up of MINIMat trial (Maternal and Infant Nutrition Interventions in Matlab, ClinicalTrials.gov identifier ISRCTN16581394). MINIMat was a factorial randomized trial with 6 arms that primarily examined the effects of food and micronutrient supplementation for pregnant women on hemoglobin level at 30 weeks’ gestation, birth weight, and infant mortality. The procedural details and results have been reported elsewhere [29,30]. A total of 4436 pregnant women were recruited between November 2001 and October 2003. This resulted in 3267 singleton live births with valid birth anthropometrics, constituting the MINIMat cohort that has been repeatedly followed up [30]. The 15-year follow-up was carried out from September 2017 to June 2019. A total of 2465 (75.5% of the eligible) adolescents completed the household survey. Over a period of 15 years some loss to follow-up was inevitable. Those with higher-educated mothers and those belonging to wealthier households were less likely to take part in subsequent follow-ups of the trial [30]. These differences, however, were small and unlikely to distort the findings. Figure 1 in the Results section presents participant flow into current study in detail.

Matlab is a rural sub-district, located about 55 km to the southeast of the capital city of Dhaka. The International Center for Diarrheal Disease Research, Bangladesh (icddr,b), formerly the South-East Asia Treaty Organization (SEATO) Cholera Research Laboratory, has been operating a Health and Demographic Surveillance System (HDSS) in Matlab since 1966. Matlab has a low-lying, deltaic topography crisscrossed by the river Gumti and its branches. Rice farming is the main occupation of people, except for a few villages that depend on fishing. A three-month long agricultural lean period usually occurs between September and November [31,32].

### 2.2. Data Collection

Trained interviewers with at least 12 years of formal schooling identified eligible adolescents using unique MINIMat identification numbers. Mother/guardian-adolescent dyads were interviewed at their residences with a pre-tested, structured questionnaire containing pre-coded questions. Insights from previous follow-ups and qualitative study undertaken at the initial phase of the 15-year follow-up [33] guided questionnaire preparation and context-specific adaptation of standard instruments. The interviewers received comprehensive training on questionnaire survey techniques and data collection using tablet computer devices. Supervisors conducted random field visits to check data collection procedure in real-time. A web portal also allowed off-site monitoring of the progress in data collection and quality of collected data.

### 2.3. Assessment of Dietary Diversity

Dietary diversity was assessed at individual level through a 24 h recall of consumed foods, using locally adapted version of a standard instrument endorsed by the Food and Agriculture Organization of the United Nations (FAO) [34]. This 10-food-group instrument has been applied widely [35,36] and validated for adolescents as well [23,37]. It offers an improvement over 2011 FAO guidelines [38] by prescribing a cut-off to differentiate adequate and inadequate DD. In our adapted version, we separated “Fish” from “Meat, poultry, and fish” to avoid masking of the plausibly lower consumption of meat by an anticipated higher consumption of fish. Whereas families in Matlab mostly buy meat from the market [33], fish remains more accessible due to its abundant natural availability in Matlab. Having fish in the same food group with meat and poultry may, therefore, obscure affordability issues and potential association with socioeconomic status. Of note, separate grouping of fish and other type of meats is also suggested in the 2011 FAO guideline [38]. We also collapsed “Pulses” with “Nuts and seeds” into the food group called “Legumes, nuts, and seeds” to better capture existing culinary norms of the setting. This grouping is also supported by the 2011 FAO guideline [38]. The context-sensitive food grouping is presented in Table 1 below.

A participant consuming at least a tablespoonful of one or more items belonging to one of the 10 food groups during the reference period received one point for that food group; otherwise, that participant received zero. Enumerators were provided with tablespoons, that they showed while ascertaining approximate consumption. To minimize recall bias, initial qualitative recall was followed by careful probing for any unreported consumption using the list of food groups and pictorial aid. This two-pronged approach has previously been recommended [39]. DDS was calculated by summing up the points for all 10 food groups- possible values ranged from 0–10. Applying the 5-food-group cut-off [34], adolescents with DDS < 5 were categorized as having inadequate DD, and those with DDS ≥ 5 were deemed having adequate DD. While the 5-food-group cut-off was prescribed for women aged 15–49 years, we reasoned that it would also suffice for boys during mid-adolescence [4]. The cut-off may not be adequate for pregnant adolescent girls in terms of micronutrient adequacy [23]. However, none of the girls in this study were pregnant.

### 2.4. Assessment of Explanatory Variables

#### 2.4.1. Socioeconomic Status (SES)

SES was derived from household asset score; a continuous, numerical variable computed by principal component analysis of data on ownership of a range of durables (e.g., mobile phone, radio, television, refrigerator, bicycle, etcetera); access to electricity and sanitary latrine; and nature of fuel used [40]. Asset scores were converted to tertiles (lower, intermediate, and upper) representing a relative measure of household SES: the poorest, the middle-status, and the richest, respectively.

#### 2.4.2. Household Food Security

We employed the Household Food Insecurity Access Scale (HFIAS) [41] to distinguish food insecure households from food secure ones. This is a 9-item, experience-based scale with a 1-month recall period and has been validated for use in LMICs [42]. In accordance with the guideline [41], food secure households were those that either had not endured any food insecurity experiences or rarely worried about running out of food. The rest were categorized as food secure.

#### 2.4.3. Maternal and Adolescent Education

Maternal education was categorized according to completed years of formal education: no education, primary (1–5 years), and secondary and above (≥6 years). Adolescents’ educational status was notably better with few of them having no forms of schooling. Rural children who enroll at public primary schools receive a monetary incentive, and this continues up to completion of secondary education. Consequently, most of the children proceed to secondary education once they have enrolled. Therefore, adolescents were categorized into two groups, primary (1–5 years) or non-formal education or illiterate and those with secondary education (6–12 years). Attending an unregistered madrasa or evening school was not considered as formal education.

#### 2.4.4. Gender and Ownership of Land

Data on gender of the adolescents (boy/girl) as well as family ownership of farming land (yes/no) and livestock (yes/no) were also collected.

### 2.5. Statistical Analysis

We adopted descriptive and inferential statistics to report the results. Sample characteristics are presented as frequency and percentage for categorical variables and as mean with standard deviation (SD) for numerical variables. Normality of DDS data and presence of disturbing outliers were examined visually by constructing histogram and boxplot. We standardized DDS by subtracting the mean from individual DDS and then dividing the result by SD. Only 3 participants (0.1% of the analytic sample) had DDS above +2.5 SD, and only 1 participant had DDS below −2.5 SD. Their removal did not impact the analysis. Difference in proportion between groups was compared with Pearson’s Chi-squared test. Means between two groups and more than two groups were tested by student’s *t*-test and one-way analysis of variance (ANOVA), respectively. All tests were two-tailed with *p*-values of less than 0.05 at 95% confidence interval (CI) considered statistically significant. A binary logistic regression model was fitted to isolate factors associated with inadequate DD. The adjusted model accounted for potential confounders selected according to existing evidence and conceptual reasoning. Crude and adjusted odds ratios (OR) with 95% CI are reported. Goodness of fit was evaluated with analysis of residuals, visual examination of the plot of quantile residuals (Appendix A) and Hosmer–Lemeshow test. Nagelkerke’s pseudo-R^2^ was computed. To evaluate gender as an effect modifier in the relationship of the explanatory variables with inadequate DD, we tested models with interaction terms incorporating gender (Appendix A). We also tested for interaction between adolescents’ education and food security or SES and found the interaction terms non-contributory. Collinearity between categorical variables was assessed by Chi-squared test, and if found dependent (*p* < 0.05), Goodman–Kruskal’s gamma (G-K gamma) value was retrieved to evaluate the strength of association. Land ownership was strongly associated with SES (*p* < 0.05, G-K gamma > 0.6) and hence not included in the final model. Strength of association of adolescents’ education with food security and SES was also within limit (G-K gamma values: −0.13 and 0.31, respectively). Variance inflation factor (VIF) for no variable exceeded 2.1. Statistical analyses were performed using the R statistical software, version 3.6.2 (Dark and Stormy Night).

### 2.6. Ethical Considerations

The Ethical Review Committee at icddr,b in Dhaka, Bangladesh, approved the 15-year follow-up of MINIMat trial. Participation was entirely voluntary. Written informed consent and assent were obtained from the participating mothers and adolescents, respectively, after full disclosure of the purpose, methods, risks, and benefits of the study as well as ensuring confidential, anonymous handling of personal information.

## 3. Results

A total of 3267 adolescents, those born as singletons with valid birth anthropometrics available, were eligible and invited to participate in the study. Amongst the invited, 2465 completed the household survey. Reasons for loss to 15-year follow-up include outmigration (n = 656), child death (*n* = 94), and refusal to participate (*n* = 52). Dietary data were missing for one participant and food security data for another one. These two were excluded from analysis, yielding an analytic sample of 2463 adolescents (Figure 1).

Table 2 demonstrates selected descriptive characteristics of the study sample. Girls constituted 51.2% of the analytic sample. While one-fifth of the mothers lacked any formal education, 80% of the adolescents had secondary education. More than half of the adolescents belonged to food insecure households. A vast majority of the households (92.9%) relied on tube-wells for drinking water. Fifty-two percent of the households owned farming land, whereas livestock ownership exceeded 70%.

### 3.1. DDS Among Adolescents and Proportion of Inadequate DD by Socio-Demographic Variables

Mean DDS and prevalence of inadequate DD across categories of socio-demographic variables are shown in Table 3. Adolescents’ mean DDS was 4.84 (SD 1.51). Small but statistically significant difference in mean DDS by gender (*p* < 0.01) and SES (*p* < 0.0001) was observed. Whereas adolescents from the poorest households had mean DDS of 4.55 (SD 1.48), their peers from the richest households had mean DDS above 5 (5.13, SD 1.56).

Nearly 58% of the adolescents consumed foods from 5 or more groups. Inadequate DD was prevalent across all socio-demographic strata. At bivariate level, proportion of adolescents with inadequate DD varied significantly by gender (*p* = 0.019), SES (*p* < 0.0001), household food security (*p* < 0.0001), and maternal education (*p* < 0.001). Proportion of inadequate DD was higher among girls than boys (44.6% versus 39.9%). Prevalence of inadequate DD followed a descending gradient from less affluent to more affluent households. More than half (50.9%) of the adolescents from the poorest households had DDS below 5 in contrast to 35.4% of their peers from the richest households. Those from food insecure households had greater proportion of inadequate DD than fellow adolescents from food secure households (47.1% versus 36.7%). No significant difference in inadequate DD was observed by educational status of the adolescents (Table 3).

### 3.2. Consumption Proportion of Different Foods and Its Socioeconomic and Gender Stratification

The proportion of adolescents consuming item(s) from the 10 food groups is reported by categories of gender and SES in Table 4. Consumption of rice (belonging to the group of grains, white roots and tubers, and plantains) was nearly universal, as only 2 participants reported not to have consumed rice. Apart from the starchy staples, consumption proportion exceeded 50% only for two food groups: fish and other vegetables. More than 70% of the adolescents consumed some type of fish. The least consumed were DGLVs (26.8%; 95% CI: 25.0, 28.5). Vitamin A-rich vegetables, tubers, and fruits were consumed by less than a third of the participants. Except fish, the consumption of animal-source foods (ASF), including milk products (30.5%), eggs (34.9%), and meat (35.2%), was noticeably low.

On stratification by gender, statistically significant difference in consumption proportion was found for five food groups. While the girls consumed more DGLVs (29% versus 24.5%, *p* = 0.011), consumption proportion was higher among boys for milk products (34% versus 27.1%, *p* < 0.001); eggs (38% versus 31.9%, *p* = 0.002); meat (38% versus 32.6%, *p* = 0.005); and legumes, nuts, and seeds (48.3% versus 44.2%, *p* = 0.038) (Table 4). Socioeconomic stratification revealed significant difference in reported consumption of five food groups: meat (*p* < 0.0001), eggs (*p* < 0.0001), other fruits (*p* < 0.0001), milk products (*p* < 0.001), and fish (*p* = 0.002). Except for fish, consumption followed a socioeconomic gradient with adolescents from the richest households consuming more than those from the poorest households. The highest difference was observed for meat—consumption being 15.3 percentage points higher among participants from the richest households than those from the poorest, followed by eggs (12.3 percentage points) (Table 4).

### 3.3. Factors Associated with Inadequate DD

The multivariable logistic regression model demonstrated that inadequate DD was significantly associated with SES, household food security, and adolescents´ and maternal education (Table 5). Compared with their peers from the richest households, the odds of inadequate DD were nearly 1.6 times higher among adolescents from the poorest households (adjusted OR 1.59; 95% CI: 1.27, 2.00). Belonging to food insecure households was positively associated with inadequate DD, as adolescents from food insecure households had 34% higher odds of inadequate DD compared to those from food secure households (aOR 1.34; 95% CI: 1.13, 1.59). In comparison to their less educated or illiterate peers, adolescents with secondary education had nearly 1.4 times higher odds of inadequate DD (aOR 1.38; 95% CI: 1.11, 1.71). Inadequate DD was significantly associated with maternal education. The odds of inadequate DD were significantly lower among adolescents of mothers with secondary education than adolescents of mothers with lower educational attainment (aOR 0.76; 95% CI: 0.60, 0.96). In the adjusted model, association of being girl with inadequate DD lost statistical significance (aOR 1.10; 95% CI: 0.93, 1.29).

Interaction terms with gender were not statistically significant, ruling out effect modification by gender (Appendix A). However, we presented stratum-specific OR in Table 5. A greater than 10% difference between adjusted OR for boys and girls was observed for food security. Controlling for the other variables, the girls from food insecure households had more than 1.4 times higher odds of inadequate DD than the girls from food secure households (aOR**_girl_** 1.42; 95% CI: 1.12, 1.81).

## 4. Discussion

The findings suggested MINIMat adolescents’ diet to be sub-optimal, characterized by a limited diversity; heavy reliance on the staple (rice); and low consumption of dark green leafy and vitamin A-rich vegetables, fruits, meat, eggs, and dairy. This resulted in a mean DDS below five (4.84 ± 1.51) with significant variation by gender and SES. Inadequate DD was present in all socio-demographic strata, and its prevalence in the cohort exceeded 40%. After controlling for confounders, belonging to the poorest households, food insecure households, and having secondary education were found to be positively associated with inadequate DD, while having mothers with secondary education or above was negatively associated with it.

Nearly universal consumption of rice found in our study is consistent with previous studies conducted in rural Bangladesh [26,43]. Such over-reliance on the staple makes the diet monotonous. This also signals policy bias at national level promoting rice production [44]. Lack of a varied diet potentially exposes these adolescents to risk of multiple micronutrient deficiencies [19] during a phase of life marked by heightened nutritional requirements. Considerably high consumption of fish probably reflects greater access of the families to different types of fish, as Matlab is a low-lying area traversed by numerous river tributaries rich in indigenous fishes. Low fruit and vegetable intake not only increases the risk of nutritional deficits but also deprives these adolescents of the benefit of lowered cardiovascular risk conferred by diets rich in fruits and vegetables [45]. Fruit and vegetable intake has been shown to decline following transition from childhood to adolescence [46]. A systematic review of 151 studies shows that diets with higher share of fruits and vegetables impart higher costs [47]. Whether the low consumption in this cohort resulted from affordability issues or lack of nutritional knowledge could not be explored within the scope of our study. Despite more than 70% of the households owning livestock, cow and goat milk producing being common in Matlab, consumption of milk products was low. While some observational studies link higher dairy consumption to health benefits like lower adiposity and higher cardiorespiratory fitness [48], qualitative exploration suggests that misconceptions regarding health benefits of dairy, dislike of the taste of milk, and peer norms may reduce adolescents’ dairy intake [49].

We documented significantly lower consumption of meat, eggs, milk products, and other fruits (a number of which are not homegrown and need to be bought from market) by adolescents from the poorest households. This aligns with existing evidence from Bangladesh [24] and elsewhere [50,51] of similar pro-rich variation and indicates the lack of purchasing power of the families in Matlab. On the other hand, contrary to Western studies that underscore gendered fruit and vegetable intake with girls consuming these more [52], we found only DGLVs, not fruits, to be consumed significantly more by girls. Gendered consumption pattern was also noted for meat, eggs, and dairy as well as legumes, nuts, and seeds: girls consumed these less. Adolescent boys in Matlab spend more time outdoors than girls, playing or engaging in strenuous activities during cultivation and harvesting seasons [33]. This could underlie intra-household inequality in food allocation that gives boys differential access [53,54] to foods perceived as capable of delivering more energy or giving bodily strength (e.g., meat, eggs, and milk). Besides, adolescent girls experience body image dissatisfaction more than boys and may be inclined toward restrictive dietary practices, even in LMIC settings [55,56]. Intentional avoidance by the girls of foods that they perceive as “fattening” (often being meat, milk, and eggs) could contribute to the gendered consumption as well [57].

Gender was not a statistically significant predictor of inadequate DD among MINIMat adolescents. A limited number of studies focused on gender-based disparity in adolescents’ DD in South-East Asia, and results are divergent. In her longitudinal study from India [58], Aurino found that boys acquired higher mean DDS at ages 5, 8, and 15 due to a significantly lower consumption of ASF, legumes, root vegetables, and fruits by the girls. This pro-boy disparity peaked at 15 years old, also the age at which we surveyed adolescents of MINIMat cohort. Nonetheless, in our adjusted model, gender was not associated with inadequate DD. Of note, Aurino used 7 food groups to assess DD, employed linear regression, and did not adjust for food security. Contrarily, another study pooling data from Ethiopia, India, Peru, and Vietnam has not found marked gender-based disparity in adolescents’ DD [59].

A strong association between SES and likelihood of inadequate DD was evident. Affordability obstacle to consuming a varied diet is well documented in LMICs, as association of higher SES with greater consumption of ASF, fruits, and vegetables appears to be robust [51]. Nutrient-rich, diverse diets cost more than diets dominated by grains and starchy staples [47,60]. Although educational status, knowledge of nutrition, taste preference, and cultural acceptability come into play, the cost dimension threatens DD the most in disadvantaged, rural settings [60]. Additionally, wealthier households may opt for using their additional income to purchase non-staple foods. Corroborating that, mothers in our qualitative study conducted in Matlab emphasized the role of the high cost of meat, eggs, milk, and certain fruits in narrowing down their options for diversifying adolescents’ diet [33].

Food security in rural areas of LMICs is continually threatened by poverty, vulnerability of traditional agriculture to extreme weather events, and escalating food prices [61]. As anticipated in a rural setting where food insecurity remains entrenched, our analysis captured a substantial association of food security (more specifically, lack of it) and inadequate DD, adding to similar findings previously reported [25,26,62,63]. Food insecurity has been shown to reduce individual-level consumption of ASF, fruits, and vegetables largely due to a significantly lower total food expenditure than food secure households [25,62,64]. The State of Food Security and Nutrition in Bangladesh 2015 reported adolescent girls to be more likely to adopt coping strategies and restrict consumption when facing food insecurity than men and under-5 children in the same household [65]. Nonetheless, we did not capture a significant difference in the odds of inadequate DD between boys and girls from the food insecure households. Blum and collaborators noted that existing gender norms limit Bangladeshi adolescent girls’ consumption of nutrient-rich foods in low-income, food insecure households [66]. Whether the influence of gender, which limits girls’ DD, becomes amplified in food insecure households remains to be explored by future studies.

We found secondary education among adolescents to be positively associated with inadequate DD. It is often postulated that schooling may equip adolescents with better understanding of what constitutes a healthy diet. However, Matlab presents a context where the school and community food environment is dominated by street food vendors selling inexpensive, nutrient-poor foods—mostly deep-fried or highly processed—that primarily attract adolescents [33]. There is no school meal program running in Matlab either. Recent studies suggest that exposure to such low-quality food environment with clustering of vendors or outlets adjacent to schools actually diminishes overall diversity of adolescents’ diet [67,68]. The association of higher maternal education and adequacy of DD agrees with previous findings from Bangladesh [27,69]. In addition to conferring empowerment, education enables mothers to understand and act upon nutrition-related information [70], plausibly contributing to a higher DD among their children. Educated mothers may also engage in income-generating activities, contributing to affordability of varied diets. According to one study conducted in Bangladesh, association of maternal education with family members’ DD varies across life span, and during adolescence, it is significantly positive for boys only [71]. Our analysis contrasted with this, showing no significant variation in association by gender.

The key strength of this study lies in the use of reliable data from an adolescent cohort in a rural area with long-established research infra-structure; high response rate; rigor in maintaining internal validity; and use of simple, cost-effective, validated tools that potentially minimized inter-rater and instrumental biases. Whereas previous studies on DD from rural Bangladesh concentrated on adolescent girls or pregnant women [23,24,27], we described and analyzed the gender dimension by including adolescent boys also. As data collection spanned two calendar years, it likely accommodated seasonal variation in consumption. It is crucial to acknowledge pertinent limitations. Cross-sectional design excluded temporality in analysis and causal inferences cannot be drawn. An array of factors influence dietary patterns and we did not incorporate all; for instance, we could not evaluate nutrition-related knowledge of the mothers and adolescents. This may have compromised the comprehensiveness of analysis. In spite of statistical modelling that accounted for multiple confounders, residual confounding could not be entirely ruled out. We ascertained DD by constructing qualitative scores (DDS) based on a single, 24-h recall, and that may not completely reflect habitual consumption, particularly for non-staples. Moreover, social desirability bias is unavoidable to some extent in self-reported dietary consumption [72]. Although one tablespoonful amount of consumption was required for scoring a point, quantifying intake in this way is difficult in a setting where family members share food from a common bowl [73].

## 5. Conclusions

This study presented evidence of limited diversity in diet in a well-characterized cohort of rural adolescents in Bangladesh. This dietary pattern is likely to place the adolescents at increased risk of nutritional deficits. SES, food security, and mothers’ and adolescents’ level of education were associated with inadequate DD. Over the past decades, nutrition of children under-five has occupied policy focus across LMICs, while adolescent nutrition remained overlooked [5], and Bangladesh has been no exception. Our findings highlighted the urgent need to invest in formulating and implementing targeted interventions to diversify rural adolescents’ diet. Intervention studies specifically addressing DD among adolescents are sparse. One Ethiopian, quasi-experimental study employing school-based, peer-led behavior change communication coupled with provision of seeds for home gardening documented a significant improvement in adolescents’ DD [74]. Furthermore, differential targeting of rural adolescents from the poorest and food insecure households is critical, as they are nutritionally more vulnerable. Targeted food value chain analysis could prove relevant in this regard [75]. Households in Matlab commonly possess some means of producing milk, eggs, and seasonal fruits. Facilitating the families’ harnessing of this intrinsic capacity and promoting homestead production can be promising [69]. In order to inform targeted interventions, future research needs to examine nutrition literacy among these adolescents and intra-household dynamics that potentially drive gendered consumption of specific foods.

## Figures and Tables

**Figure 1 nutrients-12-02230-f001:**
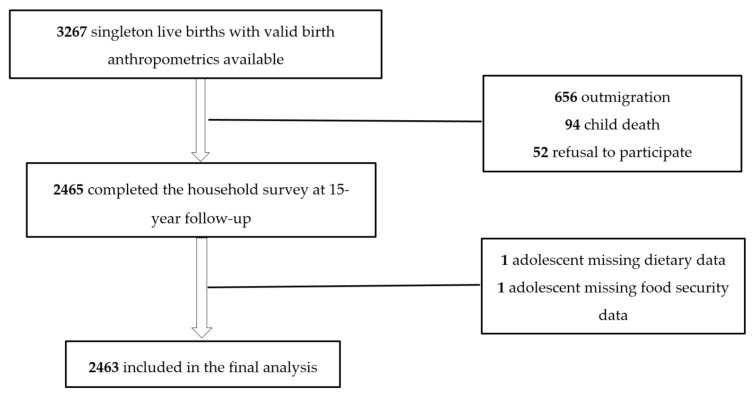
Flowchart illustrating inclusion of MINIMat (Maternal and Infant Nutrition Interventions in Matlab) adolescents into the present study.

**Table 1 nutrients-12-02230-t001:** Food items belonging to the 10 food groups used for constructing dietary diversity scores (DDS) (Bengali names are italicized).

Food Group	Individual Food Items in the Group
Grains, white roots, and tubers and plantains	Rice- cooked (*bhat*) and fried, puffed rice (*muri*), *panta*, wheat bread, paratha, chapatti, *luchi*, other items made from milled grains, maize, cassava, taro (*kochu mukhi*), green banana.
Vitamin A-rich vegetables, tubers, and fruits ^1^	Carrot, pumpkin, orange-fleshed sweet potato, mango (ripe), papaya (ripe), hog plum, watermelon.
Dark green leafy vegetables (DGLV)	Red amaranth, taro leaves, spinach, bottle guard leaves, mustard leaves, other locally available *shaak*.
Other vegetables	Tomato, gourd, brinjal, *zhinga*, long bean, cucumber, teasle gourd, wax gourd, green papaya, cabbage, cauliflower, radish.
Other fruits	Guava, banana, orange, apple, boroi, grapes, jackfruit, other fruits that are not vitamin A-rich.
Flesh and organ meat	Chicken, duck, beef, sheep, goat, pigeon, and liver, kidney, or any other organ meat.
Eggs	Chicken, duck, or quail eggs.
Fish	Rohu (*Rui*), *chitol*, *mrigal*, *shing*, *gojar*, *taki*, *puti*, tilapia, pangasius, hilsa, *kajuli*, *bashpata*, *koi*, *rani*, *bou*, dry fish, prawn, etcetera.
Legumes, nuts, and seeds	Beans, peas, lentils, hyacinth beans, pea seeds, groundnuts, peanuts.
Milk products	Milk, yoghurt, *shemai*, *shuji*, *payesh*, *khir*, paneer, or other foods made with milk.

^1^ Definition of Vitamin A-rich vegetables, tubers, and fruits was based on the FAO guideline [38].

**Table 2 nutrients-12-02230-t002:** Socio-demographic and household characteristics of the participating adolescents.

Characteristic	*n* (%) (*N* = 2463)
Gender:	
Girls	1261 (51.2)
Boys	1202 (48.8)
Adolescents’ education (completed years of formal education):	
Primary (1–5 years), non-formal, illiterate	490 (19.9)
Secondary (6–12 years)	1973 (80.1)
Maternal education (completed years of formal education):	
No education	493 (20.0)
Primary (1–5 years)	868 (35.2)
Secondary and above (≥ 6 years)	1102 (44.7)
Household food security:	
Food insecure	1327 (53.9)
Food secure	1136 (46.1)
Household source of drinking water:	
Safe sources (piped or tube-well water)	2434 (98.8)
Other sources (rain or surface water)	29 (1.2)
Household electricity coverage	2166 (87.9)
Family ownership of farming land	1290 (52.3)
Family ownership of livestock	1753 (71.2)

**Table 3 nutrients-12-02230-t003:** Mean DDS and proportion of inadequate DD across categories of socio-demographic variables.

Dietary Diversity Score (DDS)	Mean ± SD	*p*-Value ^1^
Overall	4.84 ± 1.51	--
**By gender:**		
Boys	4.92 ± 1.56	<0.01 *
Girls	4.76 ± 1.46	
**By socioeconomic status (SES):**		<0.0001 *
Poorest	4.55 ± 1.48	
Middle-status	4.84 ± 1.44	
Richest	5.13 ± 1.56	
**Proportion of inadequate DD (DDS < 5)**	*n* (%, 95% CI)	*p*-value ^2^
Overall	1042 (42.31, 40.36–44.26)	-
**Gender:**		
Boy (*n* = 1202)	480 (39.9, 37.2–42.7)	0.019 *
Girl (*n* = 1261)	562 (44.6, 41.8–47.3)	
**SES:**		
Poorest (*n* = 822) Middle-status (*n* = 819)	418 (50.9, 47.4–54.3) 333 (40.7, 37.3–44.0)	<0.0001 *
Richest (*n* = 822)	291 (35.4, 32.1–38.7)	
**Household food security:**		
Food insecure (*n* = 1327)	625 (47.1, 44.4–49.8)	<0.0001 *
Food secure (*n* = 1136)	417 (36.7, 33.9–39.5)	
**Adolescents’ education:**		
Primary, non-formal, illiterate (*n* = 490)	189 (38.6, 34.3–42.9)	0.062
Secondary (*n* = 1973)	853 (43.2, 41.0–45.4)	
**Maternal education:**		
No education (*n* = 493)	246 (49.9, 45.5–54.3)	<0.001 *
Primary (*n* = 868)	380 (43.8, 40.5–47.1)	
Secondary and above (*n* = 1102)	416 (37.7, 34.8–40.6)	

^1^ Means between two groups were tested by independent sample *t*-test and between three groups by one-way ANOVA. ^2^ Pearson’s Chi-squared test was employed to compare proportions between groups. * Asterisk indicates statistical significance at *p* < 0.05.

**Table 4 nutrients-12-02230-t004:** Proportion of adolescents who consumed item/s from the 10 food groups during the reference period by categories of gender and socioeconomic status.

Food Group		Adolescents Who Consumed Item/s from the Group *n* (% of Total in That Gender or SES Category, 95% CI)						
	Overall (*n* = 2463)	Boy (*n* = 1202)	Girl (*n* = 1261)	*p*-Value ^1^	Poorest (*n* = 822)	Middle-Status (*n* = 819)	Richest (*n* = 822)	*p*-Value ^1^
Grains, white roots and tubers, and plantains	2461 (99.9, 99.8–100)	1200 (99.8, 99.6–100)	1261 (100)	0.147	822 (100)	818 (99.9, 99.6–100)	821 (99.9, 99.6–100)	0.606
Vitamin A-rich vegetables, tubers, and fruits	774 (31.4, 29.6–33.2)	372 (30.9, 28.3–33.6)	402 (31.9, 29.3–34.5)	0.619	236 (28.7, 25.6–31.8)	262 (31.9, 28.8–35.2)	276 (33.6, 30.3–36.8)	0.095
DGLV ^2^	660 (26.8, 25.0–28.5)	294 (24.5, 22.0–26.9)	366 (29.0, 26.5–31.5)	0.011 *	218 (26.5, 23.5–29.5)	236 (28.8, 25.7–31.9)	206 (25.1, 22.1–28.0)	0.224
Other vegetables	1496 (60.7, 58.8–62.7)	724 (60.2, 57.5–62.9)	772 (61.2, 58.5–63.9)	0.616	491 (59.7, 56.4–63.1)	497 (60.7, 57.3–64.0)	508 (61.8, 58.5–65.1)	0.691
Other fruits	1112 (45.1, 43.2–47.1)	550 (45.8, 42.9–48.6)	562 (44.6, 41.8–47.3)	0.553	319 (38.8, 35.5–42.1)	391 (47.7, 44.3–51.2)	402 (48.9, 45.5–52.3)	<0.0001 *
Flesh and organ meat	868 (35.2, 33.3–37.1)	457 (38.0, 35.3–40.8)	411 (32.6, 30.0–35.2)	0.005 *	226 (27.5, 24.4–30.5)	290 (35.4, 32.1–38.7)	352 (42.8, 39.4–46.2)	<0.0001 *
Eggs	860 (34.9, 33.0–36.8)	457 (38.0, 35.3–40.8)	403 (31.9, 29.4–34.5)	0.002 *	246 (29.9, 26.8–33.1)	267 (32.6, 29.4–35.8)	347 (42.2, 38.8–45.6)	<0.0001 *
Fish	1802 (73.2, 71.4–74.9)	872 (72.5, 70.0–75.1)	930 (73.8, 71.3–76.2)	0.499	611 (74.3, 71.3–77.3)	564 (68.9, 65.7–72.0)	627 (76.3, 73.4–79.2)	0.002 *
Legumes, nuts, and seeds	1138 (46.2, 44.2–48.2)	581 (48.3, 45.5–51.2)	557 (44.2, 41.4–46.9)	0.038 *	363 (44.2, 40.8–47.6)	383 (46.8, 43.3–50.2)	392 (47.7, 44.3–51.1)	0.331
Milk products	751 (30.5, 28.7–32.3)	409 (34.0, 31.3–36.7)	342 (27.1, 24.7–29.6)	<0.001 *	207 (25.2, 22.2–28.1)	260 (31.7, 28.6–34.9)	284 (34.5, 31.3–37.8)	<0.001 *

^1^ Pearson’s Chi-squared test used to compare the consumption proportions between gender and SES categories. * Asterisk indicates statistical significance at *p* < 0.05. ^2^ Dark green leafy vegetables.

**Table 5 nutrients-12-02230-t005:** Logistic regression model analyzing association of socio-demographic variables with inadequate DD (DDS < 5).

Variables	Inadequate DD (DDS < 5)			
	Crude Analysis ^1^		Adjusted Analysis ^2,3^	
	Crude OR (95% CI) ^4^		Adjusted OR (95% CI)	
**Gender:**				
Boy (Ref.)				
Girl	1.21 (1.03–1.42) *		1.10 (0.93–1.29)	
**SES:**				
Richest (Ref.)				
Middle-status	1.25 (1.02–1.53) *		1.13 (0.92–1.40)	
Poorest	1.89 (1.55–2.30) *		1.59 (1.27–2.00) *	
**Household food security:**				
Food secure (Ref.)				
Food insecure	1.54 (1.31–1.81) *		1.34 (1.13–1.59) *	
**Adolescents’ education:**				
Primary, non-formal, illiterate (Ref.)				

Secondary	1.21 (0.99–1.49)		1.38 (1.11–1.71) *	
**Maternal education:**				
No education (Ref.)				
Primary	0.78 (0.63–0.98) *		0.85 (0.67–1.06)	
Secondary and above	0.61 (0.49–0.75) *		0.76 (0.60–0.96) *	
**Stratified analysis: by gender**				
	**Crude OR_boy_ (95% CI)**	**Adjusted ^5^ OR_boy_ (95% CI)**	**Crude OR_girl_** **(95% CI)**	**Adjusted ^5^ OR_girl_ (95% CI)**
**SES:**				
Richest (Ref.)				
Middle-status	1.33 (1.01–1.76) *	1.19 (0.89–1.60)	1.17 (0.88–1.56)	1.07 (0.79–1.44)
Poorest	1.77 (1.32–2.37) *	1.53 (1.09–2.14) *	1.93 (1.47–2.54) *	1.63 (1.20–2.22) *
**Household food security:**				
Food secure (Ref.)				
Food insecure	1.41 (1.12–1.78) *	1.26 (0.99–1.61)	1.63 (1.30–2.05) *	1.42 (1.12–1.81) *
**Adolescents’ education:**				
Primary, non-formal, illiterate (Ref.)				

Secondary	1.08 (0.83–1.40)	1.32 (1.00–1.75)	1.34 (0.96–1.87)	1.44 (1.03–2.03) *
**Maternal education:**				
No education (Ref.)				
Primary	0.85 (0.61–1.17)	0.91 (0.65–1.27)	0.72 (0.53–0.98) *	0.79 (0.58–1.09)
Secondary and above	0.61 (0.45–0.83) *	0.72 (0.51–1.02)	0.61 (0.45–0.83) *	0.79 (0.57–1.10)

^1^ Bivariable analysis of each row characteristic against inadequate DD separately. ^2^ Adjusted for gender, SES, household food security, adolescent and maternal education. ^3^ Nagelkerke’s pseudo-R^2^ value was 0.037 and *p*-value retrieved by Hosmer–Lemeshow Test was 0.17 (χ2 = 11.6, df = 8). ^4^ OR stands for odds ratio, CI for confidence interval, and asterisk indicates statistical significance as CI did not include 1. ^5^ Adjusted for SES, food security, adolescent and maternal education.

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
