# Peer review of "Exploring Rural Adolescents’ Dietary Diversity and Its Socioeconomic Correlates: A Cross-Sectional Study from Matlab, Bangladesh"

_nutrients, 2020, doi:10.3390/nu12082230_

Round 1
Reviewer 1 Report
The authors use a simple but reliable tool to assess the diet quality of a population. I have a few minor suggestions and comments.
- Abstract line 22: Please change “Belonging to the poorest” to "Among the poorest".
- Materials and methods 2.3, line 101. It is not common practice to start a sentence with an acronym. You can spell out the DD.
- Table 1, line 109: Vitamin A-rich vegetables and fruits. Please define the type of “sweet potato” listed here since not all are vitamin A-rich. Is it white or orange-flesh sweet potatoes?
- Table 1, line 109: For the “Other fruits, other vegetables” groups: Guava and tomatoes should have been listed among the vitamin A and carotenoid-rich vegetables and fruits since they are rich in beta-carotene, and as such are sources of vitamin A. Please clarify.
- Table 1, line 109: Regarding the “Other fruits”: you have defined this group as “any other fruits that are yellow/orange/red inside”. I think grapes, apple and jackfruits are misplaced because they do not have this quality of yellow/orange/red inside them. Please clarify.
- Materials and methods 2.3, line 111: What precautions or quality checks did you employ to ensure that indeed a tablespoonful was reached to score points? Please add texts to the methods section to that effect.
- There is the FVS, DDS10g and DDS (non zero), etc. In my opinion, the DDS10g could have been used for this research. Was the DDS (household measure – tablespoonful) used out of convenience?
- Results 3.2 line 226, please edit “item/s” to change to “item(s)”.
- Results 3.3 line 259-260: Simplify this phrase to be understandable to the public audience: “the odds of inadequate DD was significantly higher among adolescents with secondary education (aOR 1.38; 95% CI: 1.11, 1.71).”
- Results 3.3 line 261: “Maternal education showed an inverse association with inadequate DD”. This is double negative which does not bring out the implication. You could simply write “maternal education improved DD”.
- Results 3.3 line 262: The phrase “Adolescents of mothers with secondary education had 24% lower odds of inadequate DD versus……….”. What does this phrase mean to non-professionals and students? You can simplify this phrase to be understandable to public non-professional audience or readers. The 24% lower odds of inadequate DD = less likely to have to have inadequate DD?
- Results, Table 5 shows an odds ratio of 1.38 (1.11-1.71) for secondary education. It is not clear to me how this translates to 24% lower odds of inadequate DD versus…….. stated in line 262. Is this odds ratio in reverse order? Please clarify.
Reviewer 2 Report
The reviewer's opinion lacks the characteristics of the dietary equivalent defined by the author as (DD).
Another remark is that before deciding on the use of basic and parametric statistics, a distribution test of normality will be carried out. If so, this element should be included in the method description. It is also puzzling whether it would be worth using OR here, or whether the ANOVA regression analysis reflects the full relationship. In addition, in the reviewer's opinion - the summaries are an annual summary from which the conclusions should be clarified.
Reviewer 3 Report
The association between diet diversity and socioeconomic characteristics was investigated in Bangladeshi adolescents, and the results provided information about male adolescents, and unexpected association that secondary school education was negatively related to diet diversity. The following points should be clarified to interpret the results. The manuscript could be improved.
- Health outcomes were inconsistent in the texts. The authors mentioned in the Introduction that nutritional deficits in Bangladeshi adolescents may be risks for physical growth, physiological and cognitive development, and biological maturation (Lines 34–35), and the adolescents were stunted and/or underweight (Line 41). In the Discussion, however, low intake of some foods found in this study was possible risks for cardiovascular risk (Line 295), and adiposity and higher cardiorespiratory fitness (Line 302). The health outcomes seem in opposite direction. Discuss what health risks Bangladeshi adolescents have due to low intake the authors found in detail.
- Dietary diversity indicators have been developed for infants, young children, and reproductive women (including female adolescents), as seen in Ref. 23, 34, 35, 36, 38, etc. Explain what nutrients deficient in male adolescents, which is not include before, are critical and public concerns, and where DDS is applicable to male adolescents in the Introduction, and the Methods
- The food group that the authors used was different from the food groups in the previous reports, references 23, 35, 36, and 37. The authors divided flesh foods into fish and other meats, and combined legumes, and nuts and seeds. Explain in the Methods what of deficit nutrients the authors considered in the 10 food group the authors used.
|
Current study |
Birru Ref. 38 |
Nguyen Ref. 23 |
Arimond Ref. 35 |
Custodio Ref. 36 |
|
Countries |
Ethiopia |
Bangladesh |
Burkina Faso Mali Mozambique Bangladesh Philippines |
Burkina Faso |
|
Grains |
â—‹ |
â—‹ |
â—‹ |
â—‹ |
|
Vit. A |
â—‹ |
â—‹ |
â—‹ |
â—‹ |
|
Dark green |
â—‹ |
â—‹ |
â—‹ |
â—‹ |
|
Other vegetable |
â—‹ |
â—‹ |
â—‹ |
â—‹ |
|
Fruits |
â—‹ |
â—‹ |
â—‹ |
â—‹ |
|
Flesh, meat |
â—‹ |
â—‹ |
â—‹ |
â—‹ |
|
Fish |
||||
|
Egg |
|
|
â—‹ |
â—‹ |
|
Legumes Nuts, seeds |
â—‹ |
â—‹ |
â—‹ |
â—‹ |
|
â—‹ |
â—‹ |
â—‹ |
||
|
Milk |
â—‹ |
â—‹ |
â—‹ |
â—‹ |
- It is better to discuss whether cutoff (<5, or ≥5) is appropriate. Ref. 23 the author cited showed that cutoff, 6 foods, was appropriate instead of 5 foods in the Bangladeshi population. Or do you try cutoff 6 foods?
- Higher adolescents’ education was associated with inadequate diet diversity. Adolescents’ education was unbalanced (19.9% vs. 80.1%), though food insecurity (53.9% vs. 46.1%), and SES (822 vs. 819 vs. 822) are almost balanced. In this case, most of adolescents with lower education may be food-insecure, and had low SES; this may lead indefinite results. How about the data without food insecurity or the data without low SES in sensitivity analysis or considering interaction between adolescents’ education and food insecurity or SES.
- Is it possible to discuss whether the adolescents’ religion (Islamic, Hindu , or other) influence diet diversity?
Minor points
- The authors examined normality of DDS. “Normality of DDS data and presence of disturbing outliers were 145 examined visually by constructing histogram and boxplot.” Lines145–146. However, the result of this examination was not mentioned in the Results. The DDS has mean 4.84, where adolescents >4.84 DDS is 50% of the total, and >5 are less than 50%, but “nearly 58% of the adolescents consumed foods from 5 or more groups. Inadequate DD was prevalent across all socio-demographic strata.” Lines 215–216. This result indicates the DDS does not have a normal distribution. Desirable DDS is 10, so the distribution must not have normality. It is better to describe the results of the examination and discuss what influence the distribution has.
- The abbreviation “DDS” appeared without words at the first time in Line 106 except for the Abstract.
- Tables must be revised. Table 1 should be separated, or the layout should be changed. Table 5 has DDS <5 in the title, and DDS ≤4in the heading.
Round 2
Reviewer 3 Report
Thank you for your detailed responses; they clarified some points. But this reviewer still has some concerns.
Response to Comment 3:
“Dietary diversity was assessed at individual level through a 24-hour recall of consumed foods, using locally adapted version of a standard instrument endorsed by the Food and Agriculture Organization of the United Nations (FAO) [34]. This 10-food-group instrument has been applied widely [35,36] and validated for adolescents as well [23,37].” Lines 100–103
(1) “This 10-food group instrument” in this study is not same as the instruments in Ref. 35, 36, 23, or 37, and one of them used 9 food groups. The authors should modify explanation.
(2) Fish seems healthy rather than other meats because fish contains n3-, n6-fatty acids. The authors were likely to use the meat group in order to find the association with SES, because “families in Matlab mostly need to buy meat from the market.” The authors should explain why the meat group was separated from fish in detail in the Methods.
Responses to Comments 5 and 6:
Is it possible to add explanation described in the Responses in the Discussion?
Response to Comment 7:
Where in the Results did the author describe this assumption (or results) correspondent to examination about normality and outliers in the Methods?
